# Impact of Limited Irrigation on Fruit Quality and Ethylene Biosynthesis in Tomato: A Comprehensive Analysis of Physical, Biochemical, and Metabolomic Traits

**DOI:** 10.3390/plants14030406

**Published:** 2025-01-30

**Authors:** Nasratullah Habibi, Naoki Terada, Atsushi Sanada, Atsushi Kamata, Kaihei Koshio

**Affiliations:** 1Graduate School of Agriculture, Tokyo University of Agriculture, 1-1-1 Sakuragaoka, Setagaya-ku, Tokyo 156-8502, Japan; nt204361@nodai.ac.jp (N.T.); a3sanada@nodai.ac.jp (A.S.); koshio@nodai.ac.jp (K.K.); 2Faculty of Agriculture, Balkh University, Balkh 1701, Afghanistan; 3Department of Agriculture, Faculty of Agriculture, Tokyo University of Agriculture, Isehara Farm, 1499-1 Maehata, Sannomiya, Isehara City 259-1103, Japan; ak207913@nodai.ac.jp

**Keywords:** limited irrigation, tomato quality, ethylene production, biochemical composition, metabolome analysis

## Abstract

Water scarcity and climate change pose significant challenges to sustainable agriculture, emphasizing the need for optimized irrigation practices. This study evaluates the impact of limited irrigation (0.45 L/day per plant) compared to a control (0.87 L/day per plant) on tomato fruit quality and metabolic responses. Limited irrigation enhanced fruit flavor by reducing the pH from 4.2 to 3.4 and improved cellular integrity, with electrolyte leakage decreasing from 50% to 26%. Antioxidant levels increased, with the vitamin C content rising from 49 to 64 mg 100 g^−1^ FW, while glucose and fructose accumulation contributed to improved sweetness. Notably, limited irrigation suppressed ethylene biosynthesis, reducing methionine, ACC, ACO activity, and ethylene production, which are key regulators of ripening and senescence. This suppression suggests the potential for extending shelf life and delaying over-ripening. These findings underscore the dual benefits of limited irrigation: enhancing fruit quality and supporting sustainable water use. This research provides a viable strategy for optimizing tomato production and postharvest quality in water-limited regions.

## 1. Introduction

The tomato (*Solanum lycopersicum* L.), one of the most widely cultivated and consumed vegetables globally [1,2], is highly sensitive to environmental stresses, particularly water stress. Water stress has been extensively documented for its profound impact on plant growth and productivity [1,3]. In the context of climate change and increasing water scarcity, research on efficient irrigation strategies for tomatoes has gained significant importance [4,5]. Efficient water management is critical not only for sustaining agricultural productivity but also for conserving water resources. Optimized irrigation strategies have been shown to reduce water usage by up to 70% while enhancing crop yields by as much as 90% [6,7,8]. However, tomato productivity tends to plateau beyond a certain irrigation threshold, where additional water does not lead to further increases in yield [9,10]. To address these challenges, innovative irrigation methods and eco-friendly technologies are being explored to promote agricultural and environmental sustainability [11,12].

While achieving higher yields is vital to meet the growing consumer demand for tomatoes, excessive reliance on pesticides and fertilizers has adversely affected flavor quality [13]. Postharvest challenges, such as reduced firmness, color degradation, and flavor loss, further compromise the nutritional value and marketability of tomatoes [14]. Advances in agricultural technology have introduced sustainable practices such as substrate cultivation, hydroculture, supplementary lighting, and deficit irrigation [15,16]. Among these, deficit irrigation has gained attention for its ability to enhance nutritional and sensory attributes, including soluble solids, titratable acids, reducing sugars, vitamin C, firmness, and color [17,18,19]. However, deficit irrigation can also impair growth parameters such as plant height, leaf area, and photosynthetic efficiency [20]. Thus, identifying the optimal irrigation level is crucial to balance productivity, quality, and environmental sustainability.

Postharvest shelf life is another critical factor for tomato producers, as it significantly influences economic returns and consumer satisfaction [21,22,23]. Ethylene, a key plant hormone, plays a central role in regulating tomato ripening [24,25,26] and shelf life by influencing processes such as color development, texture softening [27,28,29], and flavor enhancement [30,31,32]. However, excessive ethylene exposure accelerates over-ripening [25,33,34], senescence [35,36], and decay [37], reducing shelf life [38] and increasing vulnerability to bruising [39] and microbial infections. Strategies such as limiting irrigation water, which slows down metabolic activity and ethylene production, and the use of ethylene absorbers have shown promise in extending postharvest shelf life. While numerous studies have investigated methods to inhibit ethylene production during the postharvest stage, there is a lack of research on the influence of preharvest practices on ethylene production.

This study hypothesizes that reducing irrigation levels will suppress ethylene biosynthesis in tomatoes, thereby slowing ripening and extending postharvest shelf life. Furthermore, we expect that limited irrigation will enhance fruit quality by increasing soluble sugars, vitamin C content, and firmness, leading to improved flavor and nutritional value.

Few studies have examined whether preharvest irrigation practices can naturally suppress ethylene biosynthesis, potentially offering a sustainable approach to extending shelf life without additional postharvest interventions. This study aims to fill this gap by investigating how limited irrigation affects ethylene biosynthesis and postharvest longevity. By assessing key biochemical markers such as methionine, ACC, ACO activity, and ethylene production, this research will determine whether water stress can slow ripening at a physiological level. Additionally, by analyzing fruit quality parameters, such as soluble sugars, vitamin C, and electrolyte leakage, this study will provide insights into the trade-offs between irrigation efficiency, fruit flavor, and storage potential.

## 2. Results

### 2.1. Physical Attributes

The findings indicate that water stress significantly altered the physical properties of the tomato fruits compared to the control. Specifically, fruit weight was reduced by 13.5% under water stress, a significant difference compared to the control group (Figure 1A). However, there was no statistically significant difference in fruit height between the two treatments, and the fruits subjected to limited irrigation treatment tended to be taller on average (Figure 1B). Similarly, while the diameter of the fruits under limited irrigation treatment was slightly smaller than that of the control, this difference was not significant (Figure 1C). In contrast, fruit volume was significantly affected, with the fruits subjected to limited irrigation treatment exhibiting a markedly smaller volume compared to the control (Figure 1D). Interestingly, this reduction in volume was accompanied by a 7% increase in fruit density under the water stress, a significant difference compared to the control (Figure 1E). Furthermore, water stress led to a substantial increase in fruit firmness, with the fruits subjected to limited irrigation treatment being 52% firmer than those in the control group (Figure 1F). These results collectively suggest that water stress impacts multiple physical attributes of tomato fruits, notably reducing size while increasing both density and firmness.

Limited irrigation treatment significantly influenced the pericarp thickness of the tomato fruits. Specifically, the tomatoes subjected to water stress exhibited a 44.96% increase in pericarp thickness compared to the fruits from the control group (Figure 2). This marked increase indicates a substantial physiological response to water stress, resulting in thicker pericarp tissue. Additionally, the pericarp of the fruits subjected to limited irrigation treatment demonstrated greater uniformity in thickness compared to the control fruits. This uniformity suggests that water stress may lead to more consistent tissue development, potentially as an adaptive mechanism to enhance fruit protection under stress conditions. These findings underscore the impact of water stress on fruit structural characteristics and highlight the adaptive physiological changes in the pericarp.

To evaluate the developmental stages of the tomato fruits under water stress treatment compared to those in the control group, we measured fruit color using a value, which indicates the level of redness. Our analysis revealed no statistically significant difference in the *a value between the water stress and control treatments, suggesting that the fruits from both conditions were at similar stages of ripeness. Despite this lack of statistical significance, there was a significant difference in the *a value: the water stress treatment group had an *a value of 20.88, while the control group had an *a value of 19.70. This corresponds to a 5.67% increase in redness for the water stress treatment, as shown in Figure 3. Although the difference was not significant, the increase in redness under the water stress conditions might reflect a physiological adaptation of the plants. This enhancement in color could provide insights into the stress responses of tomato plants and warrant further investigation into its implications for fruit quality and ripeness.

### 2.2. Ethylene Synthesis and Metabolism

Ethylene production, a key indicator of fruit quality, was significantly reduced in the fruits subjected to water stress treatments compared to the control group. Specifically, ethylene production in the control fruits was measured at 17.92 $nL\times g^{-1}\times h^{-1}$, whereas the fruits under water stress exhibited a dramatically lower production of 2.33 $nL\times g^{-1}\times h^{-1}$, representing a sevenfold decrease (Figure 4G). To further investigate this reduction in ethylene production, we examined the levels of methionine, ACC (1-aminocyclopropane-1-carboxylic acid), and ACO (aminocyclopropane-1-carboxylate oxidase), all of which are crucial components in the ethylene biosynthesis pathway. Methionine, the precursor from which ethylene biosynthesis originates, was significantly depleted in the fruits under water stress, showing a decrease of 98.66% compared to the control group (Figure 4A). This substantial reduction suggests that the initial step of the ethylene biosynthesis pathway is severely impaired under water stress conditions. ACC, the immediate precursor of ethylene, was also significantly affected, with levels being 99.10% lower in the limited irrigation treatment fruits compared to the control. This dramatic reduction further supports the observation of reduced ethylene production, as ACC availability directly limits ethylene synthesis. ACO, the enzyme responsible for catalyzing the conversion of ACC into ethylene in the final step of the biosynthesis pathway, exhibited variable activity depending on the concentration of ACC. Under limited irrigation conditions, ACO activity was significantly reduced compared to the control group. The reductions were 539.9% when the samples were kept in water, 342.4% in 1 mM ACC, 147.9% in 10 mM ACC, and 173.6% in 100 mM ACC (Figure 4C–F). Measurements were performed using gas chromatography–mass spectrometry (GC-MS) with 10 replications for each treatment. These findings suggest that water stress not only limits ethylene production by reducing ACC availability but also through the downregulation of ACO activity, further exacerbating the overall suppression of ethylene synthesis in stressed fruits.

Ethylene production in both the control and tomato fruits under limited irrigation was monitored over one month. The peak ethylene production in the control fruits occurred on day 5, reaching 18.68 $nL\times g^{-1}\times h^{-1}$, while in the fruits subjected to limited irrigation treatment, the peak was delayed until day 20, with a lower value of 8.87 $nL\times g^{-1}\times h^{-1}$. The lowest ethylene production for both treatments was recorded on the initial day and measured immediately after harvest. Throughout this study, ethylene production in the fruits under limited irrigation treatment was consistently lower than in the control. Specifically, the fruits subjected to limited irrigation treatment showed reductions of 562.6%, 658.1%, 684.9%, 178.7%, 56.5%, and 27.2% on the initial day, day 1, day 5, day 10, day 15, and day 20, respectively, compared to the control (Figure 5).

Ethylene production in both the control and tomato fruits under limited irrigation was monitored over one month. The peak ethylene production in the control fruits occurred on day 5, reaching 18.68 $nL\times g^{-1}\times h^{-1}$, while in limited irrigation treatment, the peak was delayed until day 20, with a lower value of 8.87 $nL\times g^{-1}\times h^{-1}$. The lowest ethylene production for both treatments was recorded on the initial day and measured immediately after harvest. Throughout this study, ethylene production in the fruits under limited irrigation treatment was consistently lower than in the control. Specifically, the fruits subjected to limited irrigation treatment showed reductions of 562.6%, 658.1%, 684.9%, 178.7%, 56.5%, and 27.2% on the initial day, day 1, day 5, day 10, day 15, and day 20, respectively, compared to the control (Figure 6).

### 2.3. Fruit Respiration

In the ethylene biosynthesis pathway, the conversion of ACC (1-aminocyclopropane-1-carboxylic acid) to ethylene is accompanied by the release of CO_2_ as a byproduct. An increase in CO_2_ production typically indicates elevated ethylene synthesis. However, in the present study, the results demonstrated a significant reduction in CO_2_ release in the fruits subjected to water stress compared to the control group. Specifically, CO_2_ levels decreased from 350.08 ppm in the control fruits to 280.68 ppm in the fruits under limited irrigation treatment, representing a 24.72% reduction (Figure 5B). This decline in CO_2_ release reflects the overall suppression of ethylene production in the fruits under limited irrigation treatment, consistent with the observed reductions in key ethylene biosynthesis components such as methionine, ACC, and ACO.

### 2.4. Biochemical Attributes

Biochemical parameters such as fruit juice pH, Brix, and vitamin C are critical indicators of fruit quality, influencing flavor, sweetness, and nutritional value. The pH of fruit juice directly affects its flavor profile and tends to change as the fruit ripens, making it a useful indicator of ripeness and harvest timing. In the current study, fruit juice pH ranged from 3.51 to 4.01 and was a significant 14.26% lower in the fruits subjected to water stress treatments compared to the control group (Figure 7A). This reduction in pH suggests that the fruits subjected to water stress were at a more advanced ripening stage. Glucose and fructose are sugars that affect the sweetness and overall flavor of tomatoes. Since these sugars are part of the soluble solids in the fruit, their levels play a big role in how sweet tomatoes taste. For tomatoes like Rafid, adjusting irrigation can help boost sugar levels, making the fruit sweeter and more appealing to consumers. While no statistically significant difference in Brix content was observed between the limited irrigation treatment and control fruits (Figure 7B), the Brix value was slightly higher in the fruits under limited irrigation treatment (4.84) compared to the control (4.50), indicating a marginal increase in sugar content.

Vitamin C, a potent antioxidant that protects cells from oxidative damage caused by free radicals, was significantly higher in the fruits subjected to water stress, with a 25.86% increase compared to the control (Figure 7C). This increase in vitamin C content aligns with the observed reduction in electrolyte leakage in the fruits under limited irrigation treatment, which was 57.95% lower than in the control group (Figure 7D). The reduced electrolyte leakage suggests that the fruits under limited irrigation treatment maintained better cellular integrity, further supporting the enhanced antioxidant capacity associated with higher vitamin C levels.

### 2.5. Metabolomic Analysis

In the Principal Component Analysis (PCA), the first component accounts for 48.8% of the variance, while the second component explains 23.3% of the variance (Figure 8). Together, these two components capture 72.1% of the total variance observed in the data. Metabolome profiling reveals a distinct separation between the fruits subjected to water stress treatments and the controls. Specifically, the fruits subjected to the water stress treatment exhibit higher levels of sugars, including glucose and sucrose, while the control fruits contain higher concentrations of acids such as citraconic acid, succinic acid, and fumaric acid. Additionally, the control fruits have elevated levels of citric acid and hydroxyproline compared to those subjected to the water stress treatment. An increase in hydroxyproline often signifies enhanced degradation of cell wall glycoproteins, which can result from structural breakdown. This degradation is associated with a loss of firmness and increased fruit softening, contributing to a reduced shelf life. Furthermore, the PCA indicates that the levels of methionine and ACC are higher in the control fruits than in those subjected to water stress. This suggests that ethylene production, which is influenced by these precursors, might be lower in the fruits subjected to the water stress treatment compared to the control (Figure 8).

## 3. Discussion

### 3.1. Physical Attributes

The physical attributes of the tomato fruits were influenced by limited irrigation treatment. The results showed that limited irrigation led to reductions in fruit weight, diameter, and volume. However, fruit height, density, and firmness were enhanced under limited irrigation. Meanwhile, no significant differences were observed in fruit color. Increased density could mean that the fruit is more compact, possibly indicating less air space within the fruit. This might be a desirable trait in some contexts, such as fruits meant for storage or transport, as it might reduce susceptibility to damage. Meanwhile, increased firmness is often associated with a thicker pericarp (Figure 2), which makes the fruit resistant to mechanical damage and extends its shelf life, making the fruit more suitable for handling and transportation. Research has shown that reduced fruit weight and volume often occur under water stress conditions. Studies by Farooq et al. [40] and Chaves et al. [41] found that limited water availability restricts cell enlargement and water accumulation in fruits, leading to a smaller size and lower weight. Increased density is often associated with lower water content. Martínez-Romero et al. [24] found that dense fruits tend to have improved storage qualities, including resistance to mechanical damage. Firmness is often enhanced by calcium availability and reduced irrigation. Research by Saure [42] highlighted calcium’s role in strengthening cell walls, leading to firmer fruits. Water deficit stress increases firmness, likely due to reduced cell turgor and thicker cell walls. Increased firmness of tomatoes under limited irrigation improves both consumer preferences and shipping resilience. Firmer fruits are less prone to bruising, which enhances their perceived freshness and durability. This also makes them more resilient during transportation, reducing damage and ensuring better quality upon arrival at the market. Therefore, firmer tomatoes are more suitable for commercial storage and shipping, benefiting both retailers and consumers.

### 3.2. Ethylene Biosynthesis and Metabolism

Limited irrigation treatment significantly affected fruit ethylene production, showing a substantial reduction in ethylene levels. Under limited water conditions, ethylene production decreased fivefold (3 nL g^−1^ h^−1^) compared to the control (17 nL g^−1^ h^−1^) (Figure 4G). Ethylene levels were monitored for one month at six-day intervals. Over the 30 days, ethylene production remained significantly lower in the limited irrigation treatment than in the control (Figure 5A). To investigate the mechanisms underlying this reduction, key components of the Yang cycle, including methionine, S-adenosylmethionine (SAM), 1-aminocyclopropane-1-carboxylic acid (ACC), and ACC oxidase (ACO), were measured. Methionine, the precursor in the Yang cycle, was significantly reduced by 70% under limited irrigation compared to the control. ACC oxidase (ACO) plays a crucial role in the Yang cycle, the primary pathway for ethylene biosynthesis in plants. This enzyme catalyzes the final step by converting 1-aminocyclopropane-1-carboxylic acid (ACC) into ethylene in the presence of oxygen and ascorbate, releasing carbon dioxide and hydrogen cyanide as byproducts [43]. The process begins with methionine converting into S-adenosylmethionine (SAM), which ACC synthase (ACS) then transforms into ACC, the direct precursor of ethylene. By regulating ethylene production, ACO influences fruit ripening, senescence, and stress responses. Environmental factors such as water availability, oxygen levels, and temperature significantly impact ACO activity [44], thereby affecting ethylene production. Suppressing ACO activity, as observed under limited irrigation, delays fruit ripening, extends shelf life, and enhances postharvest storage quality. Similarly, ACC levels were reduced by 55%, and ACO activity, the enzyme catalyzing ACC conversion to ethylene, decreased by 75% under limited irrigation (Figure 4A–F). This decline in ethylene production aligns with the reductions observed in the upstream metabolic precursors and enzymatic activity, emphasizing the critical role of water availability in regulating ethylene biosynthesis. Delayed ethylene production can potentially extend shelf life and reduce spoilage in commercial storage. Ethylene accelerates ripening, so by reducing its production, as seen under limited irrigation, fruit ripening slows, helping maintain fruit quality for longer. This delay in ethylene production could complement controlled atmosphere storage by keeping fruit firmer and reducing premature spoilage. Additionally, lower ethylene levels may prevent ripening from spreading to nearby fruits. However, further research is needed to assess how delayed ethylene production impacts flavor and texture during long-term storage. In addition to ethylene, fruit respiration, measured via carbon dioxide production, was also significantly reduced by 30% under limited irrigation compared to the control (Figure 5B). Reduced respiration may indicate a shift in metabolic activity under water-limited conditions, potentially contributing to the observed decrease in ethylene production. The observed reduction in respiration rates under limited irrigation suggests a shift in metabolism with practical postharvest benefits. Slower ripening, due to lower respiration and ethylene production, can extend shelf life and reduce spoilage during transport and sale. This could help preserve quality attributes like firmness and vitamin C, enhancing marketability. Additionally, the decreased ethylene production might make tomatoes less sensitive to external ethylene exposure, giving growers better control over over-ripening. These findings suggest that limited irrigation could improve postharvest management, reducing losses and optimizing fruit quality.

The findings in this study are consistent with previous research demonstrating that water stress significantly suppresses ethylene production in fruits. Fatma et al. [45] stated that under drought conditions, ACC acts as a transduction molecule for OsERF109 (ethylene response factor), which negatively regulates drought tolerance and reduces ethylene formation, acting as a switch during drought tolerance to avoid excessive ethylene production. Similarly, Fatma et al. [45] observed a decrease in ethylene production attributed to reduced enzymatic activity in the ethylene biosynthetic pathway. In contrast, Husain et al. [46] reported that while ethylene production decreased under mild water stress, severe stress conditions could trigger an increase in ethylene as a stress response hormone in many plants. This discrepancy may stem from species-specific responses, differences in water stress severity, or developmental stages of the fruit at the time of treatment. Moreover, the reduction in fruit respiration observed in this study aligns with findings from Chai et al. [47], where reduced carbon dioxide (respiration) in the leaf was linked to suppressed metabolic activity under limited water availability. The comprehensive evaluation of the Yang cycle components in this study provides mechanistic insights into how limited irrigation disrupts ethylene biosynthesis. The significant reductions in methionine, ACC, and ACO activity highlight a coordinated downregulation of the pathway, consistent with findings by [48], who demonstrated that deficit irrigation suppresses ethylene biosynthetic enzymes in African rose plums.

Limited irrigation may have a positive impact on pest and disease [49] resistance in tomatoes by enhancing the plant’s defense mechanisms. Water stress can trigger the production of certain compounds, such as secondary metabolites, that act as natural defenses against pests and pathogens. For example, limited water availability may increase the synthesis of phenolic compounds and antioxidants, which have been shown to have antimicrobial and insect-repellent properties. These compounds can strengthen the plant’s ability to resist infection and reduce susceptibility to diseases like fungal infections. Moreover, water stress may help reduce the prevalence of certain pests, as the plant’s altered chemical profile might deter them. However, the positive effects of limited irrigation on pest and disease resistance would depend on the severity of the water stress and the specific environmental conditions. More research is needed to fully understand these dynamics.

### 3.3. Biochemical Attributes

The results demonstrate that limited irrigation treatment positively affects various biochemical attributes in comparison to the control group. Specifically, the treatment significantly decreased the fruit pH from 4.2 in the control to 3.4, indicating a more acidic environment, which could enhance flavor or improve fruit quality. This reduction in pH may be linked to the way limited water stress influences the biochemical composition of the fruit, promoting the production of organic acids. In addition to the pH change, electrolyte leakage was significantly lower in the limited irrigation treatment (26%) compared to the control (50%), suggesting that the treatment helps maintain better cellular structure and integrity, reducing cell damage and preserving fruit quality during storage and handling. Moreover, the limited irrigation treatment led to an increase in vitamin C content, rising from 49 mg 100 g^−1^ FW in the control to 64 mg 100 g^−1^ FW. This increase suggests that water stress triggers the plant’s defense mechanisms, promoting the synthesis of antioxidants and thereby enhancing the nutritional value of the fruit. Overall, the limited irrigation treatment not only improves fruit quality by lowering the pH and increasing the vitamin C content, but it also enhances cellular integrity, highlighting its potential for optimizing water usage while maintaining or even improving fruit quality. From a marketability standpoint, fruits with a higher vitamin C content can command premium prices, particularly in markets where consumers are increasingly aware of the health benefits of antioxidants. Additionally, with the growing consumer preference for functional foods—those that offer additional health benefits beyond basic nutrition—tomatoes with a higher vitamin C content could be marketed as a value-added product. This can lead to increased sales and market differentiation, benefiting producers who can leverage these health benefits in their marketing strategies. Furthermore, in the context of a more competitive marketplace, such quality improvements could help position the product as a preferred choice among health-conscious buyers, contributing to brand loyalty and customer retention.

Several studies have reported that water stress can result in increased acidity in fruits, potentially due to the accumulation of organic acids such as citric or malic acid, which are often promoted under limited water conditions. Lu et al. [50] observed that reduced water availability in tomatoes resulted in a lower fruit pH and an increased organic acid content, similar to the findings in this study. This could be explained by the plant’s response to stress, where metabolic pathways shift to preserve cellular functions, resulting in the accumulation of acids (Figure 8). The lower electrolyte leakage observed in this study (26%) under limited irrigation also mirrors findings from other studies, indicating better cellular integrity under water stress. ME et al. [51] found that limited irrigation improved membrane stability in tomatoes, as evidenced by a reduction in electrolyte leakage. This indicates that even limited irrigation induces stress. The increase in vitamin C content observed in this study (64 mg 100 g^−1^ FW in limited irrigation vs. 49 mg 100 g^−1^ FW in control) is also consistent with the results of other research indicating that water stress can enhance antioxidant production. Medyouni et al. [52] reported that reduced irrigation significantly increased the vitamin C content in tomatoes, as water stress stimulates the synthesis of antioxidants, including vitamin C, as part of the plant’s defense mechanism against oxidative damage.

### 3.4. Metabolomic Analysis

The metabolome analysis highlights significant biochemical changes in fruit composition under limited irrigation. Increased glucose and fructose levels suggest osmotic adjustment, as plants accumulate sugars to maintain turgor under water stress. These sugars also enhance fruit sweetness and flavor, improving marketability. Conversely, lower methionine and its derivative, ACC, indicate suppressed ethylene biosynthesis, potentially delaying ripening and extending shelf life. In contrast, the fruits subjected to the control treatment had higher methionine and ACC levels but lower sugar content, suggesting that water availability accelerates ripening at the expense of sweetness. These findings demonstrate the dual benefits of limited irrigation: enhancing flavor quality through sugar accumulation and improving postharvest stability by reducing ethylene-related processes. This underscores the potential of targeted irrigation strategies to optimize fruit quality and sustainability.

This study’s findings align with existing research on fruit responses to limited irrigation while offering unique insights. The increase in glucose and fructose under limited irrigation supports studies like Tao et al. [53], which highlight sugar accumulation and quality improvement in apples under water stress as a mechanism to enhance sweetness and flavor. The reduction in methionine and ACC complements findings by Fatma et al. [45], linking water stress to suppressed ethylene biosynthesis, delayed ripening, and improved shelf life. Notably, this is the first report of a combined increase in sugars with reduced methionine and ACC under limited irrigation, providing novel insights into the interplay of metabolic and hormonal pathways enhancing fruit quality and storage potential.

#### Implications

The findings of this study highlight several practical implications for tomato production, particularly in water-scarce regions or markets prioritizing quality. Limited irrigation enhances key physical attributes, such as increased firmness and density, which improve storage, transportation, and shelf life, although at the cost of reduced fruit weight and size. The suppression of ethylene production and reduced respiration rates further extend postharvest stability, allowing better alignment with market demand. Biochemical improvements, including a higher vitamin C content, lower pH, and increased glucose and fructose levels, enhance nutritional value, flavor, and consumer appeal, providing a competitive edge in premium markets. These changes underscore the potential of deficit irrigation as a sustainable practice that optimizes water usage while maintaining or improving fruit quality. Additionally, insights into the metabolic and hormonal adjustments observed under water stress offer valuable guidance for breeding programs that develop drought-tolerant, high-quality tomato varieties. By integrating water management strategies with market and breeding priorities, farmers can enhance production efficiency, reduce environmental impact, and meet evolving consumer preferences.

While this study provides valuable insights into the effects of limited irrigation on tomato quality and shelf life, it is important to note some limitations. The research was conducted over a single growing season, and the long-term effects of limited irrigation on fruit growth, yield, and postharvest attributes remain uncertain. A multi-season study would provide a more comprehensive understanding of how seasonal variability and environmental factors might influence the outcomes.

## 4. Materials and Methods

### 4.1. Plant Materials and Treatments Application

The experiment was conducted at Green Farm Lapin, Namegawa-cho, Saitama Prefecture, Japan, using the Rafid tomato cultivar. The ‘Rafid’ tomato variety was selected for this study due to its large fruit size, juiciness, and high market demand. Known for its climacteric ripening process, ‘Rafid’ is sensitive to ethylene regulation, making it ideal for examining how limited irrigation influences ripening and shelf life. This variety’s rich flavor, balanced acidity, and high sugar content are highly valued in the market, and optimizing irrigation strategies can improve fruit quality while reducing water usage. Two irrigation treatments were implemented: normal irrigation and limited irrigation. In the normal irrigation treatment, seeds were sown on 4 August 2023 and transplanted on 31 August 2023, and fruit harvesting commenced on 3 November 2023, with an average irrigation amount of 0.87 L/day per plant. In the limited irrigation treatment, seeds were sown on 5 September 2023 and transplanted on 4 October 2023, and fruit harvesting began on 4 December 2023, with an average irrigation amount of 0.45 L/day per plant. Stone wool cube was used as the cultivation media. In this study, drip irrigation was used to ensure consistent water distribution and precise control over the water stress levels across all treatments. Drip irrigation allowed for a targeted application of water directly to the root zone, minimizing wastage and ensuring uniform water delivery. The leaf water potential (Ψ_1_) was measured using a pressure chamber to assess the degree of water stress in the plants. The results showed that the control group maintained a Ψ_1_ of −0.5 MPa, indicating no water stress, while the limited irrigation group exhibited a Ψ_1_ of −1.2 MPa, reflecting moderate stress. Drip irrigation effectively maintained consistent water stress conditions across treatments, ensuring accurate comparisons and reliable results. Both treatments were carried out in separate greenhouses maintained at a daytime temperature of 30 ± 4 °C and a nighttime temperature of 15 ± 4 °C. Rockwool was used as the growing medium, and irrigation was provided via a drip irrigation system using a nutrient solution. The nutrient solution was prepared with the following concentrations: nitrogen (21.85 mg/L), phosphorus (13.7 mg/L), potassium (46.35 mg/L), magnesium (8.8 mg/L), iron (0.357 mg/L), manganese (0.038 mg/L), boron (0.058 mg/L), copper (0.004 mg/L), zinc (0.007 mg/L), and molybdenum (0.002 mg/L). Details of the irrigation treatments and room temperature were recorded on a daily basis and are illustrated in Figure 9A,B.

The fruits from the control and those subjected to limited irrigation treatments were harvested and transported to the Laboratory of Tropical Horticulture Science at Tokyo University of Agriculture for quality analysis. Ten fruits from each treatment were randomly selected for each measurement to ensure accurate and representative data.

### 4.2. Physical Attributes Measurement

Fruit weight was measured from 10 fruits, each treatment using a digital balance and displayed as grams (g). Fruit height was measured using a digital vernier caliper and displayed as millimeters (mm). Fruit diameter was also measured using a digital vernier caliper and displayed as millimeters (mm). Fruit volume was measured using a 500 mL beaker filled with water. The fruits were put inside the beaker, and after taking them out, the amount reduced was counted as fruit volume, which was displayed as mL. Fruit density was measured using the equation $D=m\times v^{-1}$, where D represents density, m stands for mass or weight, and v stands for volume. Fruit density was presented as $g\times {cm}^{-3}$. Fruit firmness was checked using a penetrometer (Instron 3342, Illinois Tool Works Inc., Hopkinton, MA, USA) from 10 fruits for each treatment, and the data were expressed as N [54]. Pericarp thickness was measured using a ruler and was expressed as millimeters (mm). In this experiment, pigmentation was measured using Colorimeter NR-3000 [55].

### 4.3. Ethylene Measurement

For the measurement of ethylene, all the tomato fruits were harvested at the same stage of ripeness and the same size and color. Then, fruits were weighed, and the initial ethylene production was measured. The tomato fruits were enclosed in 550 mL jars for one hour under a black sheet of cloth to prevent the light from affecting them; 1 mL of gas was extracted using a plastic syringe. The gaseous sample was injected into the GC-FID (gas chromatography–flame ionization detector) (GC-14B), and the data were expressed as nL × g^−1^ × h^−1^ for further analysis [56,57,58].

### 4.4. Quantification of ACO

Disks of fruit tissues, measuring 5 mm in diameter, were taken out with a cork borer; four disks were taken from slices of three fruits per treatment and placed in ice cube trays. Then, 5 mL drops of water (control), 1 mM, 10 mM, and 100 mM ACC were applied to each of the disks. The disks were saturated with ACC solution and soaked for 1 h. Finally, the disks were transferred into bottles and then incubated (25 °C) for 1 h. Later, 1 mL of gas from the headspace was withdrawn for analysis of the ethylene contents. Ethylene was detected using gas chromatography, specifically GC-FID (gas chromatography–flame ionization detector) (Agilent Technologies Inc., Santa Clara, CA, USA), and data were expressed as nL g^−1^ h^−1^.

### 4.5. Quantification of ACC

Methionine and ACC were measured using GC-MS, which is well-detailed in metabolome analysis.

### 4.6. CO_2_ Release Measurement

For the measurement of carbon dioxide, the fruits were weighed, and then the CO_2_ was measured. The fruits were enclosed in 550 mL jars for 1 h under a back sheet of cloth to prevent the light from affecting them; 1 mL of gas was extracted using a plastic syringe. The gaseous sample was injected into the GC-TCD (gas chromatography–thermal conductivity detector) (Agilent Technologies Inc., CA, USA), and the data were expressed in µL × g^−1^ × h^−1^ for further analysis [59].

### 4.7. Quantification of Fruit Biochemical Attributes and Metabolomic Analysis

The selection of biochemical and metabolic traits for this study was based on their relevance to assessing tomato fruit quality and the impact of limited irrigation on fruit development. Fruit juice pH was chosen as it directly relates to flavor, with a lower pH often indicating enhanced acidity and improved taste. Brix (soluble solids content) is an important measure of sweetness, reflecting the concentration of soluble sugars in the fruit, which are crucial for consumer preference. Vitamin C content, a key antioxidant, was included to evaluate the nutritional benefits and shelf life of tomatoes, as increased levels are associated with enhanced fruit quality. Electrolyte leakage was assessed to gauge the integrity of fruit cells under water stress, as lower leakage indicates better cellular health and reduced damage. Lastly, metabolomic analysis provided a comprehensive view of the fruit’s biochemical composition, identifying metabolic changes such as increased sugar content, which influence flavor and quality. These traits collectively offer a detailed understanding of how limited irrigation affects both the quality and nutritional value of tomatoes, providing insights into sustainable irrigation practices for optimal fruit production.

Fruit juice pH was measured using a pH meter (LAQUAtwin-pH-11; Horiba Ltd., Kyoto, Japan). Brix was recorded using a Brix meter (Hybrid PAL-BX I ACID F5; Atago Co., Ltd., Saitama, Japan). Vitamin C (ascorbic acid mg/100 g) was measured using a reflectometer (RQflex plus; Merck, Darmstadt, Germany) and ascorbic acid strips (Reflectoquant^®^; Merck). First, 1 g of pulp was mixed with 2 mL of metaphosphoric acid at 5% in a 1.5 mL Eppendorf tube; then 1 mL of the mixture was centrifuged at 25 °C and 5000 rpm for 5 min using a centrifuge (MX-307; Tomy Seiko Co., Ltd., Tokyo, Japan); finally, the test strip was immersed in the solution and placed in the reflectometer. Electrolyte leakage, indicative of fruit damage, was assessed by determining the number of electrons leaking from the fruits. Ten random fruits were selected for measuring electrolyte leakage in each treatment, treating each fruit as a replication. Fruit cuttings, made using a 1 cm diameter stainless steel cork borer, were stored in pure water in 2 mL tubes at room temperature (25 ± 1 °C) for half an hour. The electric conductivity (EC) of these fruit cuttings was measured using an electrical conductivity meter (LAQUATWIN-S070, Horiba Scientific Ltd., Kyoto, Japan) [60].

Metabolomic analysis was conducted following the method previously explained by Oliver [61] with some modifications. Ten replicated samples of individual fruits from each treatment were homogenized using pre-cooled mortars and pestles with liquid nitrogen. A hundred grams of the resulting puree was used for the extraction. Methanol (250 µL) and one zirconia bead were added to each sample and mixed well using a tissue layer (Oscillating Mill MM 400, Retsch GmbH, Haan, Germany) at 27 Hz for 2 min. After adding 250 µL of chloroform, samples were put in a thermo mixer (Eppendorf Thermomixer F2.0, Hamburg, Germany) for 3 min at 37 °C, 1200 rpm. The standard solution of 50 µL and 175 µL of ultra-pure water were subsequently added to the mixture and centrifuged (TOMY MX-307 high-speed refrigerated microcentrifuge, Tokyo, Japan) at 120 × 100 rpm for 10 min at 25 °C. The standard solution was prepared by diluting 0.2 mg of Ribitol in 1 mL of ultra-pure water. Then carefully, 80 µL of supernatant from each sample was added to the 1.5 mL of Eppendorf tube and put in a centrifugal vaporizer (EYELA CVE-200D, TOKYO RIKAKIKAI Co., Ltd., Tokyo, Japan) for 2 h with a cooling trap apparatus (EYELA UT-80, TOKYO RIKAKIKAI Co., Ltd., Tokyo, Japan). After that, samples were transferred to a freeze dryer (EYELA FDM-1000, TOKYO RIKAKIKAI Co., Ltd., Tokyo, Japan) and kept overnight. The resulting residues were dissolved in 40 µL of Methoxyamine hydrochloride solution by putting in the thermo mixer for 90 min at 37 °C followed by adding 50 µL of N-Methyl-N-trimethylsilyl tri fluoroacetamide (MSTFA) with another 30 min incubation in the thermomixer under the same condition. Then 50 µL from the extraction was used to analyze metabolomics components. The Methoxyamine hydrochloride solution was prepared by diluting 20 mg of Methoxyamine hydrochloride in 1 mL of Pyridine.

Metabolomic analysis was performed using gas chromatography–mass spectrometry (GC-MS-QP2010 plus, SHIMADZU, Tokyo, Japan). The column used was DB-5 (0.25 mm internal diameter, 30 cm length, and 1.00 µm of film thickness, Agilent Technologies Inc., Santa Clara, CA, USA). The GC conditions were as follows: The oven temperature was held for 1 min at 60 °C, raised to 320 °C at a rate of 4 °C min^−1^, and held at 10 min, with the flow rate of Helium 1.1 mL min^−1^. The analysis method of mass spectrometry was scan mode and conditions. The transfer line was set at 290 °C, and the ion source was kept at 200 °C. Mass spectra were recorded at a scan s^−1^ with an m z^−1^ 45–600 scanning range.

### 4.8. Statistical Analysis

We used R software (version 4.1.2) to analyze the data. To determine whether there were significant differences between the treatment groups, we used a one-way analysis of variance (ANOVA) with a significance level set at *p* < 0.05. This means that we looked for differences in the results that were unlikely to have occurred by chance. Additionally, we used Principal Component Analysis (PCA) to explore the key metabolites responsible for the observed differences, helping us understand the main factors influencing the data.

## 5. Conclusions

This study reveals that limited irrigation enhances fruit quality by modifying key biochemical and metabolic pathways. Reducing the pH and increased organic acid accumulation improve flavor, while enhanced membrane stability and lower electrolyte leakage contribute to better cellular integrity. The rise in vitamin C content under water stress boosts the fruit’s nutritional value and marketability. Metabolomic analysis indicates that increased glucose and fructose enhance sweetness, while reduced methionine and ACC suppress ethylene biosynthesis, delaying ripening and extending shelf life. These findings suggest that limited irrigation acts as a physiological regulator, optimizing fruit resilience and quality while reducing water usage. By integrating biochemical and metabolomic insights, this study provides a theoretical framework for targeted irrigation strategies that enhance crop quality and sustainability. Future research should investigate the molecular mechanisms governing these adaptations to refine irrigation practices and improve agricultural efficiency in water-limited environments. This research emphasizes the role of limited irrigation in sustainable agriculture, offering an innovative approach to improving postharvest quality while reducing water consumption. Future research should focus on the long-term impacts of limited irrigation on yield and the genetic mechanisms involved, helping further develop eco-friendly and resource-efficient farming practices.

## Figures and Tables

**Figure 1 plants-14-00406-f001:**
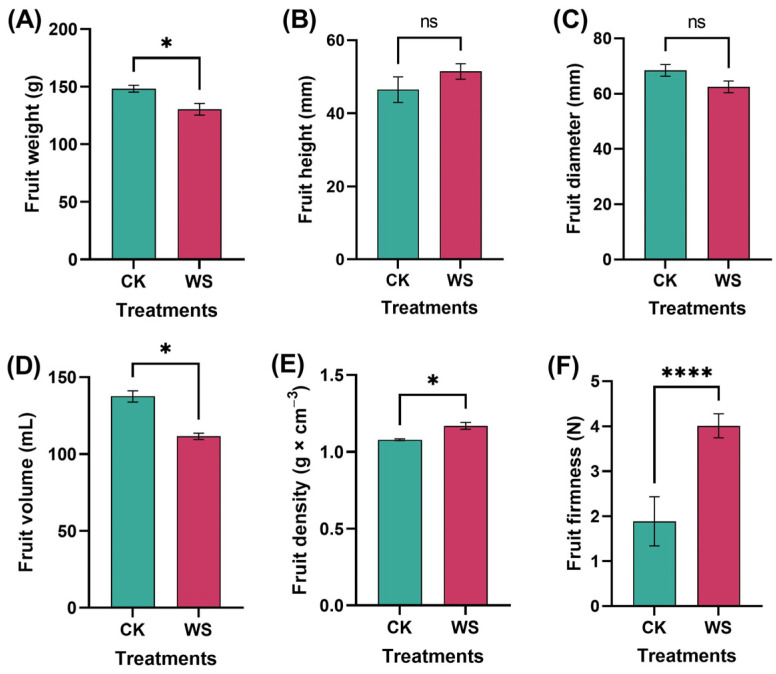
Comparison of physical parameters between tomato fruits grown under normal and limited irrigation conditions, including fruit weight (**A**), fruit height (**B**), fruit diameter (**C**), fruit volume (**D**), fruit density (**E**), and fruit firmness (**F**). CK: control, and WS: water stress. The data, presented as mean ± SD, represent the average of 10 individual fruits (*n* = 10) for each condition. Statistical significance is indicated by ns (not significant), * *p* < 0.05, and **** *p* < 0.0001.

**Figure 2 plants-14-00406-f002:**
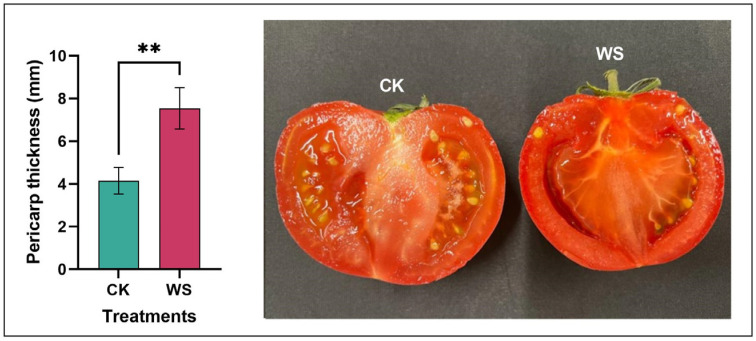
Comparison of pericarp thickness in tomato fruits under normal irrigation (control) and limited irrigation (water stress) conditions. The data, presented as mean ± SD, represent the average of 10 individual fruits (*n* = 10) per condition. Statistical significance is indicated by ** *p* < 0.01. This analysis highlights the impact of irrigation levels on the structural characteristics of the tomato pericarp, which may be crucial for fruit quality and marketability.

**Figure 3 plants-14-00406-f003:**
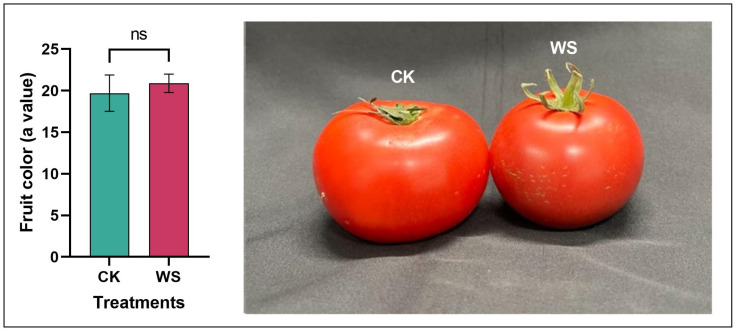
Comparison of fruit color (*a value) in tomato fruits under normal irrigation (control) and limited irrigation (water stress) conditions. The data, presented as mean ± SD, represent the average of 10 individual fruits (*n* = 10) per condition. Statistical significance is indicated by ns (not significant). This analysis examines the effects of irrigation levels on the color quality of tomatoes, which can be a key factor in consumer preferences and marketability.

**Figure 4 plants-14-00406-f004:**
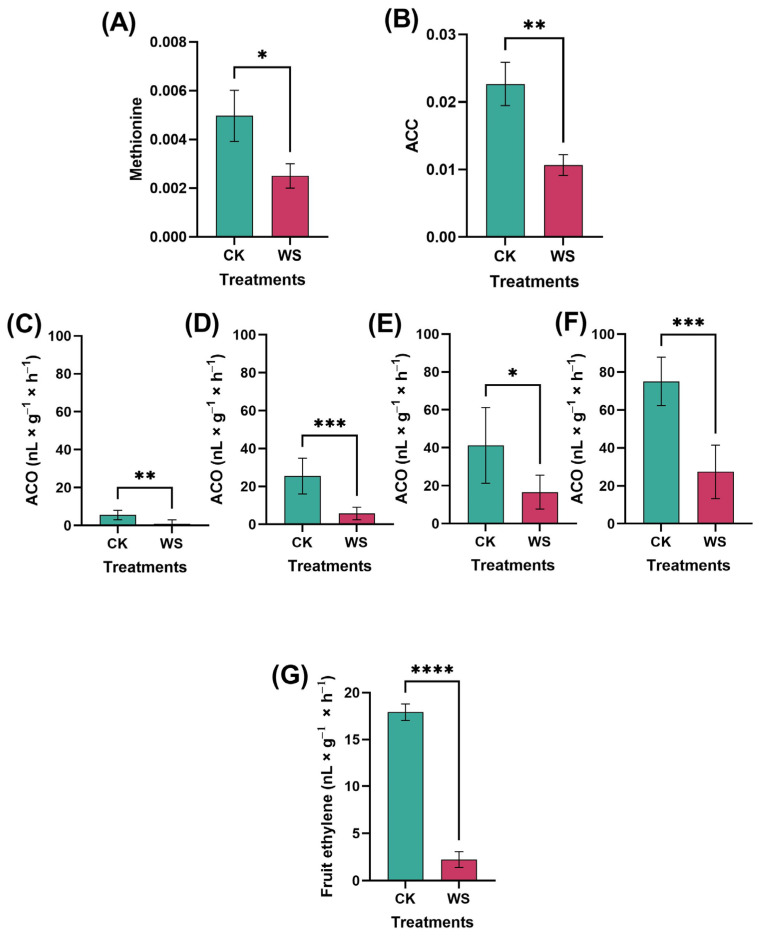
Ethylene-related parameters in tomato fruits under normal (control) and limited irrigation (water stress) conditions: methionine (**A**), ACC (**B**), ACO activity in water (**C**), ACO activity in 1 mM ACC (**D**), ACO activity in 10 mM ACC (**E**), ACO activity in 100 mM ACC (**F**), and ethylene production (**G**). The data, presented as mean ± SD, represent the average of 10 individual fruits (*n* = 10) for each condition. Statistical significance is indicated by * *p* < 0.05, ** *p* < 0.01, *** *p* < 0.001, and **** *p* < 0.0001. This analysis investigates the impact of limited irrigation on key ethylene-related parameters, which influence ripening and postharvest quality in tomatoes.

**Figure 5 plants-14-00406-f005:**
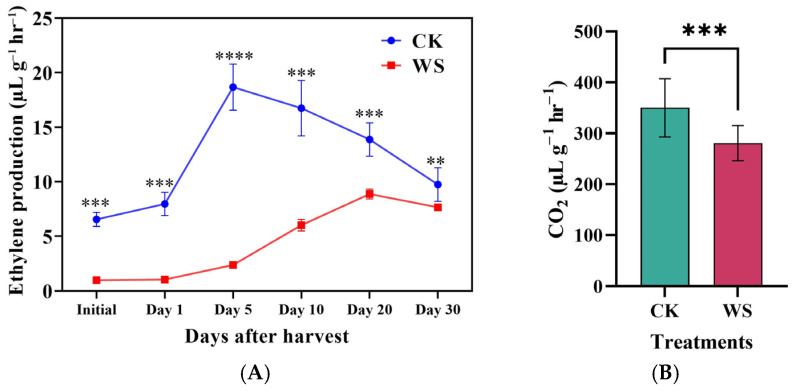
Ethylene production over one month (**A**) and average carbon dioxide production (**B**) in tomato fruits under control (CK) and limited irrigation (WS) conditions. The data are presented as mean ± SD, with 10 individual fruits (*n* = 10) per treatment. Statistical significance is indicated by ** *p* < 0.01, *** *p* < 0.001, and **** *p* < 0.0001. This analysis explores how limited irrigation affects ethylene and carbon dioxide production, two key parameters related to fruit ripening and postharvest behavior.

**Figure 6 plants-14-00406-f006:**
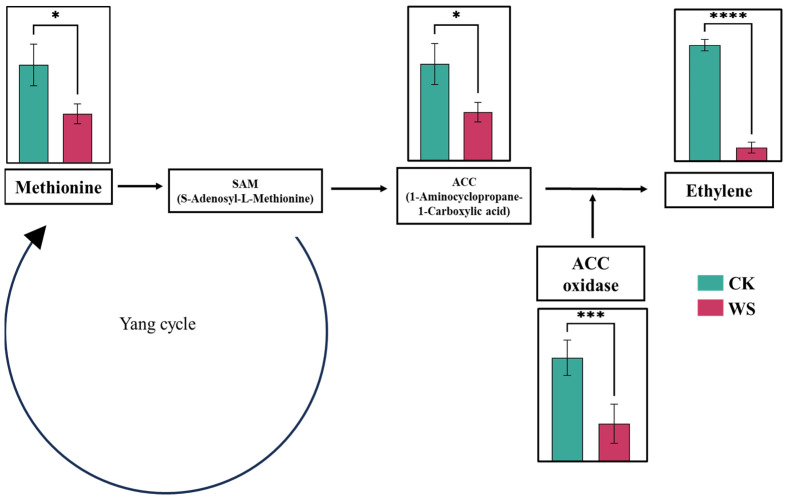
Effects of limited irrigation on ethylene biosynthesis in tomato fruits. Ethylene-related parameters measured include methionine, ACC, ACO in water, ACO in 1 mM ACC, ACO in 10 mM ACC, ACO in 100 mM ACC, and ethylene production under control (CK) and limited irrigation (WS) conditions. The data are presented as mean ± SD, with 10 individual fruits (*n* = 10) per treatment. Statistical significance is indicated by * *p* < 0.05, *** *p* < 0.001, and **** *p* < 0.0001. This analysis highlights how limited irrigation influences key ethylene biosynthesis parameters, potentially affecting fruit ripening and quality.

**Figure 7 plants-14-00406-f007:**
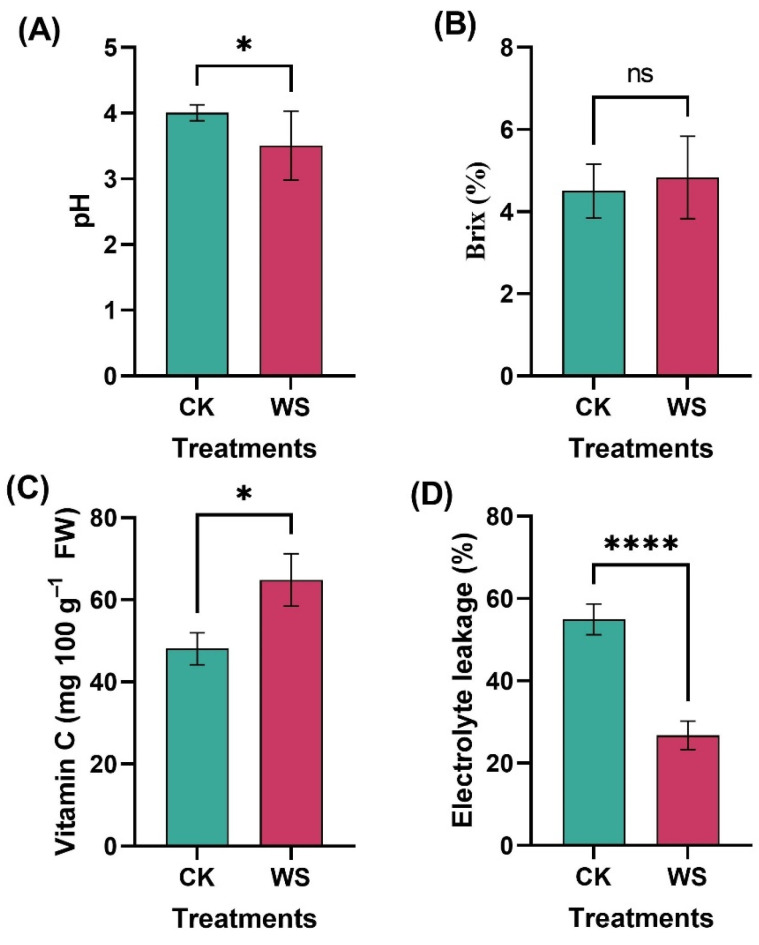
Biochemical traits of tomato fruits under normal and limited irrigation conditions: pH (**A**), Brix (**B**), vitamin C content (**C**), and electrolyte leakage (**D**). CK: control, and WS: limited irrigation treatment. The data, presented as mean ± SD, represent the average of 10 individual fruits (*n* = 10) for each condition. Statistical significance is indicated by ns (not significant) and * *p* < 0.05, **** *p* < 0.0001. This figure illustrates the impact of limited irrigation on key biochemical traits, highlighting potential changes in fruit quality and nutritional value.

**Figure 8 plants-14-00406-f008:**
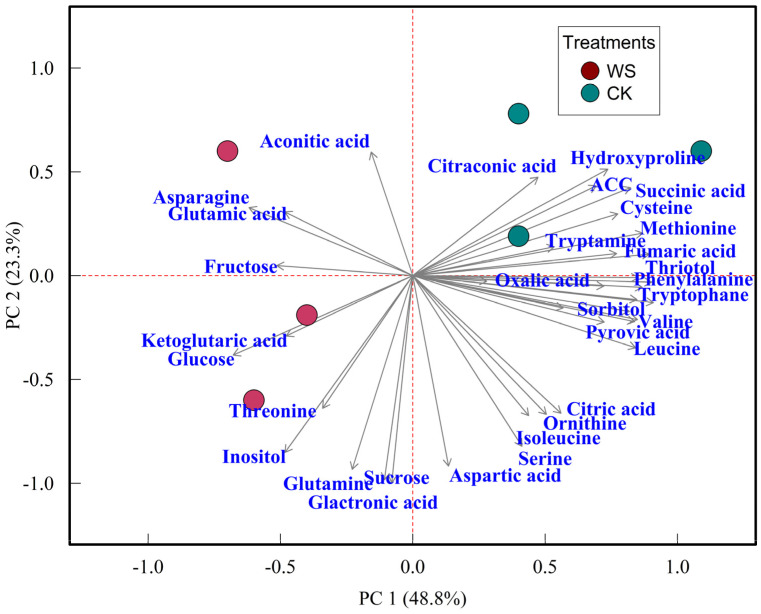
Principal Component Analysis (PCA) of metabolite profiles in tomato fruits under control and limited irrigation treatments. WS: limited irrigation, and CK: control. The PCA was performed to explore the differences in metabolite composition between the two treatment groups, providing insights into the metabolic changes induced by water stress. Each point represents a sample, and the clustering of data points reflects the similarity in metabolite profiles.

**Figure 9 plants-14-00406-f009:**
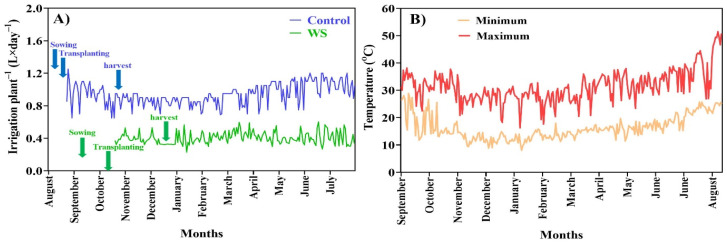
Irrigation per plant in both normal and limited irrigation treatments (**A**) and minimum and maximum room temperature (**B**). WS: limited irrigation.

## Data Availability

Data will be available upon request from the corresponding author.

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
