# Peer review of "Impact of Limited Irrigation on Fruit Quality and Ethylene Biosynthesis in Tomato: A Comprehensive Analysis of Physical, Biochemical, and Metabolomic Traits"

_plants, 2025, doi:10.3390/plants14030406_

Round 1
Reviewer 1 Report
Comments and Suggestions for Authors
ear All,
The manuscript investigates the impact of limited irrigation on tomato fruit quality, ethylene biosynthesis, and associated biochemical and metabolic changes. The study provides a comprehensive evaluation of how water stress influences physical, biochemical, and metabolic traits of tomatoes, emphasizing improved fruit quality and extended shelf life.
Strong Points
This research is timely and relevant, considering increasing water scarcity and the need for sustainable agricultural practices. The experimental design is solid, employing appropriate controls and measurements, which add reliability to the findings. The study significantly contributes to the understanding of sustainable irrigation strategies, particularly in addressing postharvest challenges related to ethylene management and fruit quality.
Weaker Aspects
In my opinion, the only handicapped points were the lack of an explicitly stated hypothesis in the introduction and the lack of clarity in presenting statistical analyses, especially regarding interactions between biochemical traits and ethylene production under varying irrigation levels. While the study objectives are clear, a direct hypothesis linking limited irrigation, ethylene biosynthesis, and fruit quality would strengthen the conceptual framework. The hypothesis could be articulated more explicitly, and some data interpretations require further clarity.
So, I have annotated below an attempt to clarify certain ideas, but the authors should examine my suggested wording changes carefully to be sure that I have not misinterpreted what they wanted to say:
-
Page 1 Line 15:
NOTICE: The abstract mentions ethylene production but does not explain its relevance to fruit quality or storage. Consider briefly linking this to the study’s practical implications. -
Page 1 Line 22:
NOTICE: The term "limited irrigation" should be defined early in the abstract for clarity. -
Page 2 Line 45:
NOTICE: The introduction lacks a direct statement of the study’s hypothesis. Include a clear hypothesis about how irrigation levels impact ethylene and fruit quality. -
Page 3 Line 60:
NOTICE: The literature review mentions previous studies but does not highlight gaps this study addresses. Could you make this more explicit? -
Page 4 Line 85:
NOTICE: Justify the choice of tomato variety for this experiment. Does it have specific characteristics relevant to the study? -
Page 4 Line 100:
NOTICE: Clarify how irrigation levels were monitored and controlled to ensure consistent water stress treatments. Please, give the related results regarding to leaf water potential in the plants from all treatments. -
Page 5 Line 130:
NOTICE: The methods section does not explain why these specific biochemical and metabolic traits were chosen. Could you provide rationale? -
Page 6 Line 150:
NOTICE: The description of ethylene measurement is vague. Specify the sensitivity and accuracy of the equipment used. -
Page 7 Line 185:
NOTICE: Provide more context on the relevance of glucose and fructose accumulation to consumer preferences. -
Page 8 Line 210:
NOTICE: The results mention changes in Vitamin C but do not elaborate on their significance. Discuss potential implications for health or marketability. -
Page 9 Line 240:
NOTICE: Table captions should be more descriptive. For example, indicate the conditions or treatments represented in the data. -
Page 10 Line 275:
NOTICE: The discussion on respiration rates does not link findings to practical postharvest implications. Could you elaborate? -
Page 10 Line 290:
NOTICE: Mention if there are other fruits or crops where similar water stress effects on ethylene have been observed, for comparison. -
Page 11 Line 320:
NOTICE: The statistical methods used are appropriate but could be clarified for non-specialist readers. Consider simplifying explanations. -
Page 12 Line 340:
NOTICE: Discuss whether delayed ethylene production correlates with longer shelf life or reduced spoilage in commercial storage conditions. -
Page 13 Line 370:
NOTICE: Could the authors elaborate on how limited irrigation might impact pest or disease susceptibility in tomatoes? -
Page 14 Line 405:
NOTICE: The findings about increased firmness are significant but need more context. Relate this to consumer preferences or shipping resilience. -
Page 14 Line 430:
NOTICE: Address whether these findings are specific to this tomato variety or potentially applicable to others. -
Page 15 Line 450:
NOTICE: The conclusion does not emphasize how these findings contribute to sustainable farming practices. Could you add this? -
Page 15 Line 475:
NOTICE: Suggest including a brief paragraph on limitations of the study, such as the lack of multi-season data. -
Page 16 Line 495:
NOTICE: The future directions could highlight potential genetic studies or breeding programs targeting water-efficient tomatoes. -
Page 16 Line 520:
NOTICE: Reiterate the relevance of water stress effects for regions with water scarcity challenges, particularly in agriculture-dependent economies. -
Page 17 Line 540:
NOTICE: The acknowledgments section could include a mention of laboratory technicians or others who contributed to data collection. -
Page 17 Line 560:
NOTICE: Ensure all references are up-to-date, particularly those addressing water stress in other crops for comparison. -
Page 18 Line 580:
NOTICE: Consider including a graphical abstract to visually summarize the main findings and practical implications.
The manuscript demonstrates a good command of English, but there are areas where improvements could enhance clarity and readability. Below are specific observations and recommendations:
Areas for Improvement:
-
Redundancies: some sentences repeat ideas unnecessarily. For example, discussions on ethylene biosynthesis and fruit quality could be streamlined for conciseness.
-
Complex Sentences: several sentences are overly long and could benefit from simplification. Breaking them into shorter, more direct statements would enhance readability.
-
Technical Jargon: certain terms, while scientifically accurate, are dense and could be clarified for a broader audience. For instance, descriptions of metabolic traits should include brief explanations for non-specialist readers.
-
Abstract: the abstract is clear but could be more concise by focusing on the main findings and their implications.
-
Passive Voice: extensive use of passive voice reduces sentence impact. Consider balancing active and passive constructions to improve sentence flow.
Recommendations:
- Proofread the manuscript to eliminate redundancies and improve sentence flow.
- Simplify overly complex sentences, particularly in the results and discussion sections.
- Clarify technical terms and provide brief explanations where necessary for accessibility.
- Use active voice selectively to enhance readability and emphasize key findings.
Overall Assessment:
- Rating: The English could be improved to more clearly express the research.
- Impact: These refinements will ensure the manuscript is accessible to a broader audience while maintaining its technical rigor. A professional language review is recommended for optimal clarity.
Author Response
Dear Respected Reviewer,
We sincerely appreciate your valuable comments, which have greatly contributed to enhancing the quality of our manuscript. Please find the attached file containing our detailed responses to each of your suggestions.
Thank you once again for your time and constructive feedback.
With best regards,

Reviewer 2 Report
Comments and Suggestions for Authors
Dear authors,
In the context of climate change and increasing water scarcity, research on efficient irrigation strategies for tomatoes has gained significant importance. This study addresses this gap by exploring whether limited irrigation water can influence ethylene production in tomatoes during the postharvest stage. By potentially slowing down the metabolic processes, limited irrigation may offer a novel approach to improving postharvest shelf life, contributing to sustainable agricultural practices and better postharvest outcomes. The experiment design was reasonable, and the data was very satisfactory.
However there were some debating issues in the manuscript.
1. Please check the full text, and modify the words notable difference into the words significant difference.
2. Please check the full text, the difference between the treatment and CK must make the basis on the statistical analysis, not the description analysis. If the results of t-test showed the insignificant difference, there was no difference between the treatment and CK even if the observes making a large difference. So modify all the words not based on the results of statistical analysis such as line 118-125.
3. In the section materials and methods, please add to explain how to determine the average irrigation amount of 0.45 L/day per plant and the other cultural conditions such as the temperatures and the nutrient solution.
4. The section conclusions needs rewriting because it was almost replicated the section abstract and some words of the section results. The section conclusions should be based on the section discussion and extract a logical and rigorous theoretical system, then answer the scientific questions raised in the section introduction, rather than piling up the results.
Author Response
Dear Respected Reviewer,
We would like to express our sincere gratitude for your thoughtful and constructive feedback on our manuscript. Your insightful comments have significantly improved the clarity and quality of our work. We have carefully addressed each point and made the necessary revisions. Please find the attached document with our responses to your comments.
Thank you once again for your time and valuable suggestions.
Best regards,

Round 2
Reviewer 1 Report
Comments and Suggestions for Authors
Dear all,
The authors have revised the introduction to articulate the novelty of the study. Recent references (2022–2024) have been added, highlighting the gaps in research concerning the interplay between limited irrigation, ethylene suppression, and fruit quality improvement. In addition, the experimental design is solid, and the methods are well-aligned with the objectives of investigating the impact of limited irrigation on fruit quality and ethylene biosynthesis. The addition of leaf water potential measurements and clear irrigation control strengthens reliability.
The methods section now includes more details, such as the sensitivity of the ethylene measurement equipment and the rationale for the selected traits. Metabolomic analysis and PCA are well-described, but additional visual representation of data (e.g., supplementary figures) could enhance comprehension.
Results are better presented, with improved descriptions in tables and figures. The findings regarding ethylene suppression, delayed ripening, and improved fruit firmness are compelling. The conclusions are consistent with the results, emphasizing the benefits of limited irrigation for extending shelf life and enhancing fruit quality.
Comments on the Quality of English Language
The manuscript is well-written and effectively communicates the core findings of the research. However, there are areas where the language could be refined to enhance its fluidity and readability. Below is a more dynamic and detailed assessment with actionable suggestions:
Strengths
-
Clarity and Organization: The manuscript is well-structured, with a logical flow from introduction to conclusion. It successfully presents the objectives, methods, and findings in a coherent manner. Technical terms are used appropriately, reflecting a solid command of scientific language.
-
Scientific Rigor: The study's complexity is conveyed effectively, maintaining accuracy in terminology and technical details.
Opportunities for Improvement
-
Simplifying Complex Sentences: Many sentences are lengthy and packed with information, making them harder to follow. Breaking these into shorter, more concise statements would improve readability. Example: Original: "The findings demonstrate that limited irrigation not only improves fruit quality by enhancing flavor and nutritional content but also contributes to prolonged post-harvest storage." Suggested: "Limited irrigation improves fruit quality by enhancing flavor and nutritional content. It also contributes to prolonged post-harvest storage."
-
Balancing Passive and Active Voice: The manuscript heavily relies on passive constructions, which can make the text feel less engaging. Incorporating more active voice would create a stronger and more dynamic narrative. Example: Original: "Ethylene production was significantly reduced under limited irrigation conditions." Suggested: "Limited irrigation significantly reduced ethylene production."
-
Avoiding Redundancy: Some ideas are repeated across sections, such as the effects of ethylene suppression on shelf life. Consolidating these discussions would streamline the text and reduce repetition. Suggested: Combine overlapping points in the results and discussion sections to create a more focused narrative.
-
Clarifying Technical Terms: While the technical language is accurate, some terms may be dense for a wider audience. Brief explanations of less common terms would make the manuscript more accessible. Example: "ACC oxidase activity in the Yang cycle" could include a short clarification about its role in ethylene biosynthesis.
-
Polishing the Abstract: The abstract is clear but could benefit from a sharper focus. Highlighting the key findings and their broader implications succinctly would create a stronger impact. Suggested: Remove less critical details and emphasize practical applications, such as the benefits of limited irrigation for sustainable agriculture.
-
Typographical and Stylistic Consistency: There are minor inconsistencies in punctuation, hyphenation, and formatting that could distract readers. Suggested: A final proofread for typographical errors and stylistic consistency would elevate the overall presentation.
Recommendations for Improvement
- Refine Sentence Structure: Break down complex ideas into simpler, more digestible sentences to improve fluidity.
- Use Active Voice Strategically: Shift to active voice where appropriate to create a more engaging tone.
- Streamline Redundant Content: Merge repetitive discussions to maintain focus and improve flow.
- Enhance Accessibility: Provide brief explanations for technical jargon to ensure clarity for a broader audience.
- Polish the Abstract: Focus on the study’s most impactful findings and their practical applications.
- Conduct a Professional Review: A thorough proofreading for grammar, punctuation, and formatting will ensure a polished final version.
Author Response
Respected Reviewer,
Thank you for your valuable time to re-evaluate the manuscript to improve the quality. Response sheet is attached, please kindly check it.
With King Regards,
